# OpenReview forum: "Hidden in Plain Text: Emergence & Mitigation of Steganographic Collusion in LLMs"
_ICLR.cc/2025/Conference — Submitted to ICLR 2025_

### Official Review · Reviewer_vrcM · 2024-10-30

**Soundness:** 3
**Presentation:** 4
**Contribution:** 3
**Rating:** 6
**Confidence:** 4

**Summary:**

This paper investigates the emergence and mitigation of steganographic collusion. The studied scenario is reinforcement learning with high optimization pressure, where two LLM agents are implicitly forced to learn to communicate biased/secret information via steganography. Specifically, the term steganography refers to the behavior that two LLMs (learn to) encode and decode biased/secret information subtly, e.g., by using different synonyms, which is hard to be perceived or detected by another LLM. Under this setting, this paper proposes a general method for eliciting steganography in frontier LLMs, especially in the presence of an overseer, and shows that current mitigation methods are not robust to the learned steganography scheme.

**Strengths:**

### Originality
* The idea of studying steganographic collusions in LLMs *under optimization pressure* is novel to my knowledge.
* The related works and contributions over prior works are sufficiently discussed. There is a close prior work studying a similar setting; this work does not leverage end-to-end gradients to optimize the collision.

### Quality
* The methodology is well-defined and solid. The two learning environments GBRL and ICRL are also clearly defined.
* It's appreciated that the paper has conducted very comprehensive evaluations.

### Clarity
* The paper is well-written and easy to follow. All terms are clearly defined.

### Significance
* The overall idea that AI systems, especially LLM agents, can learn to behave in unintended and hard-to-detect ways is interesting and useful.

**Weaknesses:**

### Originality
I don't have concerns in this perspective.

### Quality

**Q1: Regarding the proxy for optimization pressures.**

The main claim in this paper is that optimization pressure can force the LLMs to collude via stenography. It's unclear why adding an overseer LLM could be regarded as a proxy for the optimization pressure (if I understand correctly). Given that the presented datasets are biased by nature (also see Q3), the presence of an overseer would only encourage the biased information to be encoded in a less perceivable way, instead of encouraging the emergence of stenography.

**Q2: The mitigation strategy is limited to individual messages.**

As discussed in Q3, the stenography is more or less a result of overfitting. For example, the biased information about good/bad universities is encoded by using different synonyms when summarizing the cover letter. This is different from traditional stenography, where it could be easier to get detected in a single message. In this case, however, it seems one can collect multiple messages from the LLMs and check if a pattern exists that potentially encodes biased information. For the sake of detecting unintended behavior in LLM agents, conceptually, one can analyze multiple summaries and identify that pattern.

### Clarity
I don't have concerns in this perspective.

### Significance

**Q3: Results are expected given that the datasets are biased by nature.**

In Section 4, several biased-by-nature datasets are used to conduct the learning task without any techniques for mitigating overfitting. In other words, the learner is forced (and expected) to overfit the 100% biased dataset, yet the "non-trivial takeaway" in this paper is more or less saying that the LLMs can indeed learn to communicate that biased information. It's just the presence of an overseer will encourage the resulting communication to be more innocuous-looking. For example, the Uni dataset explicitly assigns the ground-truth label based on biased information (good university vs. bad university); there seem to be no outcomes other than overfitting.

Note that I'm not saying the setting is wrong or not meaningful -- the overall framework and evaluations are still impactful for investigating stenography between LLM agents, under the assumption that stenography is expected. What I'm concerned is that the "emergence of stenography" is a somewhat trivial conclusion based on the setting. For example, what is the difference between the "emergence of stenography" and the "emergence of overfitting"?

**Questions:**

See weaknesses.

---

> ### Author Response · Authors · 2024-11-19
>
> We thank the reviewer for their comprehensive review and positive comments. We’re glad you find the overall idea novel, interesting and useful, and appreciate that you find the related work sufficient, the methodology solid, and the evaluations comprehensive.
>
> ### Q1: Regarding the “proxy for optimization pressure”
>
> The overseer is not a proxy for optimisation pressure, it is one part of the two components of the optimization pressure:
> * Reward from collusion (e.g due to biased datasets in the environment)
> * Punishment for collusion being spotted by the overseer
>
> It is necessary for both of these factors to be present for collusion via steganography to emerge, and both factors are likely to occur in many real-world scenarios. The overseer is part of the environment that shapes the optimization landscape. We demonstrate that under these specific pressures, steganography can emerge as an instrumental goal even when not explicitly programmed.
>
> For example, in the candidate screening setting, the agents need to correctly classify candidates based on cover letters while minimizing penalties from the overseer (for revealing bias). The biased nature of the dataset creates a scenario where perfect performance is only achievable by exploiting the bias. The overseer, by penalizing explicit use of the bias, forces the models to find alternative ways to achieve their goal, that is by leveraging steganographic collusion.
>
> ICRL and GBRL are the two training algorithms in which context the reward and punishment are applied. These methods serve as proxies for the kinds of optimization that developers may apply.
>
> ### Q2: The mitigation strategy is limited to individual messages
>
> Note that the overseer in the ICRL experiments (O2 overseer) is shown a batch of messages, and asked to evaluate them all together, so it does have the context of multiple messages. For GBRL, evaluating multiple messages simultaneously was not feasible due to GPU memory constraints. In this way, our results highlight the computational tax that engineers of the real-world systems we emulate would need to pay in order to get robust mitigation of steganography. Our constraints mirror those real-world constraints.
>
> However, we agree with your point that stronger overseers could be developed. Perhaps by asking a model to search for correlations between biasing information and the CV summaries. But it is worth noting that without ground-truth labels, or even prior knowledge of the semantics of the steganography, this is a difficult task beyond the scope of our paper.
>
> Ultimately, we used overseer strategies that we thought might reasonably be applied in practice, as we were interested in if steganography would arise without being explicitly optimized for in a realistic setting. We believe that in many scenarios in which developers would train a system to perform some task with constraints (do not collude), developers may use a fixed overseer. In line with assumptions made in previous research [2], we did deliberately make the overseer ‘weaker’ than the colluding agents. The assumption in [2] is that a weak system may be easier to guarantee the behavior of, therefore could be expected to reliably act in accordance with the developers goals. However, our findings show that in this setting, the colluding team will exploit these weaknesses.
>
> ### Q3: Results are expected given that the datasets are biased by nature
>
> To ensure that the communication strategies learned were not overfit to the training samples, we used train/test splits and reported results on the test sets. Additionally, in the decentralized learning settings the agents were trained on different splits.
>
> We agree the models are performing well by exploiting a misspecified reward function: to do well the LLMs have to adopt steganography. However, it was an open question whether models would be able to learn this strategy or not.
>
> Previous work by Motwani et al [1] has shown that linguistic stego is a non-trivial capability for today’s frontier LLMs. In their paper, they showed that the GPT-4 class of models was able to successfully perform stego less than 30% of the time **even when prompted to do so**.
>
> We were curious if LLMs could learn to perform steganography when their environment encourages them to do so (but without explicit prompting). We came up with a simplified version of a scenario that could occur in the wild, as most datasets include harmful biases. In this scenario, during LLM training, due to a misspecified reward that encourages an unintended behavior, and inadequate oversight, the LLM learns to execute the unintended behavior via steganography such that the overseer cannot detect it.
>
> [1] Motwani, Sumeet Ramesh, et al. Secret Collusion Among Generative AI Agents https://arxiv.org/abs/2402.07510 (2024)
>
> [2] Ryan Greenblatt, et al. AI Control: Improving Safety Despite Intentional Subversion https://arxiv.org/abs/2312.06942 (2024)

---

> > ### Comment · Reviewer_vrcM · 2024-11-21
> >
> > Thanks for the response. I would like to focus on Q3 here.
> >
> > I think this is the same concern as Reviewer qY8i although I'm talking through it from the overfitting perspective. The author's response is not very convincing in addressing the fundamental question here -- **for the given setting, there is no outcome other than encoding the biased information using other tokens, which is stego.** The reason is very simple, if the learning algorithm is provided with a 100% biased dataset and forced to fit the dataset, it will eventually overfit to the biased distribution.
> >
> > The authors cited Motwani et al [1] to justify the hardness of making LLMs generating stego. I think this is the right direction where it's necessary to establish a baseline that stego is not expected. However, the cited paper uses prompts instead of a learning algorithm to elicit stego. This is not sufficient for a baseline in this paper because there is no optimization in place and the LLMs may not be able to follow the prompt that easily.
> >
> > I'm also confused about the response where the authors use train/test split to avoid overfitting. First of all, train/test split is not the way to mitigate overfitting. Common textbook techniques would include dropout, regularization, early stopping, etc. Since the train/test splits have the same biased distribution, once the model overfits to the training set, as encouraged by the learning algorithms and settings, it's almost impossible that the model will not learn the biased information.
> >
> > From this perspective, what this paper shows (as an analogy to classification problems), is that when given a 100% biased dataset (where the ground-truth label is decided only by the biased information), it's surprising for the learning algorithm (under the optimization pressure) to generate a model that makes the prediction based on the biased information.
> >
> > For the emergence of stego to be recognized as "surprising", I think it would be necessary to show that under a similar optimization setting, it's very hard for LLMs to come up with a collusion stego. But this does not seem to be possible, because again, the setting is designed in a way such that stego / overfitting is the only option -- you cannot expect a non-overfitting result if the problem forces the learning algorithm to overfit.
> >
> > To summarize, I think the evaluations in this paper are good examples of how to learn stego strategies in RL & LLMs, but are not sufficient to show that such observation is surprising, mainly because the problem setting is somewhat self-contradicting. I would encourage the authors to think about what optimization scenarios make it "hard" for the stego to emerge, so that the emergence shown in this paper is unexpected. My initial thought is to apply mechanisms like early stopping or regularization, but that would immediately contradict the "optimization pressure" assumption. I feel that the proposed notion of "optimization pressure" and the outcome of stego are similar in spirit to the interplay between regularization and overfitting.

---

> ### Author Response · Authors · 2024-11-22
>
> We thank the reviewer for their perspective and engagement with our response.
>
> We agree it is not overly surprising that steganography emerged in our settings. Indeed, these settings were explicitly designed to encourage such emergence.
>
> However, we emphasize that (i) we do not claim the results are surprising in our paper, and (ii) whether or not they are surprising is not essential to any of our three main contributions.
>
> **Contribution 1:** As the reviewer points out, our elicitation methods “are good examples of how to learn stego strategies in RL & LLMs”.
>
> **Contribution 2:** Though the natural emergence results may not be overly surprising, they empirically demonstrate that, concerningly, unintended stego could occur in realistic settings (where there is a misspecified reward and inadequate oversight) in LLMs. This is a valuable contribution which highlights the need for further research into studying the emergence of stego in LLMs and designing improved mitigation methods.
>
> We agree with the reviewer that investigating stego emergence in different problem settings, including ones where it seems harder for the stego to emerge, is a very interesting direction. We hope our initial results can serve to encourage and inform further research down these directions.
>
> **Contribution 3:** We show that current methods for mitigating steganography in LLMs are often not robust to optimization pressure (i.e., this contribution is orthogonal to whether our natural emergence results were surprising).

---

> > ### Comment · Reviewer_vrcM · 2024-11-25
> >
> > Thanks for the response, I think I should clarify that my intention is to help qualify the significance aspect of this paper. I'm generally okay with the other three aspects, including the originality. However, the significance aspect is more important in this case and outweighs the originality aspect if not adequately addressed IMO.
> >
> > The current motivation and presentation of this paper primarily focus on the "emergence of stego", as indicated by the title, intro, etc., which gives a clear impression that this is to some extent a surprising finding. From this perspective, I think the current presentation is a bit overclaimed, as the paper's setting was designed such that stego has to emerge. Moreover, given that similar settings (using RL with other models instead of LLMs) have been studied before, there is no clear reason to set a prior expectation that such settings, when transiting to LLMs, would not work.
> >
> > Based on my understanding, a more precise description of this work is "evaluating learned-stego using RL & LLMs and the robustness of mitigation strategies". Any points beyond this scope do not necessarily qualify the significance aspect of this paper:
> > 1. Does the finding that LLMs can be directed to learn stego give some new insights? Not exactly, because prior studies using RL & other models have already shown the possibility.
> > 2. Does this finding raise any brand new unexpected concerns? Not exactly, the setting is too optimal to raise a concern. People from stego and/or ML backgrounds would both agree that this finding is expected. Some extension on this point -- it might be beneficial to have a setting where the dataset is not biased and enforce a similar level of "optimization pressure". This would strengthen the understanding of such a concern -- what would happen if we have a more practical-looking scenario.
> > 3. Does the results improve our understanding of the performance of how LLMs can be directed to learn stego? Yes, as most of the evaluations have answered the question of, to what extent can LLMs be directed to learn stego when given a biased dataset.
> >
> > I would thus recommend tuning down the paper as a plain evaluation instead of saying that this is a new insight and brand-new concern. The current presentation is far less sufficient to support the latter claim.

---

> > > ### Author Response · Authors · 2024-11-25
> > >
> > > Thank you for the response. These are valuable points and we hope they can help us clarify and improve the presentation of our paper.
> > >
> > > First, we will explicitly establish what we believe to be the significance of each of our contributions (including reframing our thoughts based on the reviewers feedback). (We apologize for any duplication in restating Contributions. We restate here to ensure we have explicitly argued for their *significance*).
> > >
> > > ### _Significance_ of Contribution 2:
> > >
> > > We agree that given the setup, many would not be surprised that stego has arised. Our arguments for why this contribution *is nevertheless significant* is as follows.
> > >
> > > **(1) Our result can serve as an empirical demonstration that can draw attention to, and encourage further research into, the risk of unintended stego emerging as an instrumental goal in LLMs.**
> > > While previous work has briefly touched on this risk, this has not yet been done in depth, and there have yet to be any empirical demonstrations of the risk occurring in LLMs. We believe instrumental emergence of stego in LLMs is a concerning risk. As such, we believe it is valuable for a paper to highlights this risk *front-and-center*, providing concrete empirical results that demonstrate the possibility of this risk, and discouraging the design and use of reward signals that can inadvertently encourage stego – no prior research can serve this purpose.
> > >
> > > **(2) Our setting, though extreme, bears on key aspects of concerning scenarios where stego may unintentionally emerge in realistic situations.**
> > > Unintended stego emerges in our setting due to: (i) a mispecified reward (in our case derived from a biased dataset) that encourages a misaligned/unsafe behaviour, and (ii) oversight which discourages obvious execution of this behaviour. These are key aspects that may lead to the emergence of stego in practice. For example, [1] present the example of a model being rewarded for being sycophantic, but being punished for being blatant in the reasoning that produces the sycophantic behaviour.
> > >
> > > Our setting is a somewhat extreme case – stego is strongly encouraged by the reward signal. However, it does show that 'unintended' stego can emerge due to flaws in the reward design. We would argue this presents initial evidence that, under enough optimization pressure, stego could also emerge when the reward component inadvertently encouraging stego is less substantial (though future research should investigate this further).
> > >
> > > We would be eager to hear the extent to which the reviewer agrees/disagree with the above points. **We will rewrite relevant parts of the paper tomorrow** (allowing some time, in case the reviewer wishes to respond) t**o better present our work according to these points and the reviewers suggestions**, and to ensure we are not making inaccurate claims regarding the surprising-ness of our findings.
> > >
> > > As a side-note, the following suggest our results in Section 5.1 may not be fully unsuprising:
> > > - Prior work has struggled to elicit stego in LLMs via ‘realistic’ GBRL [2]. We are, to the best of our knowledge, the first to achieve this.
> > > - We note the ICLR results with Claude were surprising. Whilst a model will refuse to propagate biases in the datasets if asked explicitly to do so, we found that ICRL (with no weight changes) was enough to elicit this behaviour in Claude 3.5-sonnet. Note this was not true in all models we tested.
> > > - We were unsure whether the stego would be successful when we added the paraphraser. We assumed this would make stego much harder - as shown in previous literature [1]. Yet, both GBRL and ICRL successfully bypassed the paraphraser in Section 5.1, showing it is ‘surprisingly’ hard to prevent emergence of stego in LLMs with standard mitigation methods.
> > >
> > >
> > > ### _Significance_ of Contribution 1:
> > > We present practically useful, general methods for automatic elicitation of stego in LLMs. This is significant as these methods can facilitate several important research avenues.
> > >
> > > ### _Significance_ of Contribution 3:
> > > We show stego defenses are not always robust. This is significant as it motivates the need for improved defences.
> > >
> > > [1] Roger, Fabien, and Ryan Greenblatt. "Preventing Language Models From Hiding Their Reasoning." arXiv preprint arXiv:2310.18512 (2023).
> > >
> > > [2] Some negative steganography results. https://www.lesswrong.com/posts/EEvsL9cpgDAxAhTzt/some-negative-steganography-results

---

> > > > ### Comment · Reviewer_vrcM · 2024-11-29
> > > >
> > > > Thanks for the response. I think the main disagreement here is "*unintended* stego emerging".
> > > >
> > > > The presented emergence of stego relies on two factors:
> > > > 1. A biased dataset that, when given to a reasonably good learner, it is almost impossible for the learner to not fit with 100% accuracy. I'm pretty sure even a linear classifier would be able to learn the manually encoded biased information because this is how the ground-truth label is designed.
> > > > 2. A reward (and the corresponding learning environment) that, the only foreseeable outcome is stego. (ref Reviewer qY8i)
> > > >
> > > > I'm curious how this setting would support the claim of "unintended" -- the setting is so extreme that everything is intended. It may be true that the reward did not explicitly encourage the learner to literally learn a stego strategy, but the only feasible outcome for the learner is that the manually encoded biased information will be learned, and learned in a way that the biased information is encoded using some different tokens. The only thing special about this outcome, is that the notion of "encoding biased information using other tokens" is stego.
> > > >
> > > > Given the above two factors, I think for the "unintended" claim to stand, at least from a dataset perspective, the paper would probably need to change its settings by using an unbiased dataset, where (1) the ground-truth label is not determined based on the biased information, and (2) there are some potentially misuseable features presented, such as gender, university goodness. Then, evaluate the emergence of stego under different levels of optimization pressure. This is a more precise alignment of "unintended stego" in real-world settings.

---

> ### Author Response · Authors · 2024-12-01
>
> We thank the reviewer for their response, and are glad we are converging to the crux of our disagreement.
>
> ## We explicitly care about settings where the reward is mispecified and encourages a bad behaviour.
>
> >I think for the "unintended" claim to stand, at least from a dataset perspective, the paper would probably need to change its settings by using an unbiased dataset, where (1) the ground-truth label is not determined based on the biased information
>
> Perhaps this points to another key disagreement/misunderstanding. The threat model we are concerned about (a misspecified reward combined with an overseer, leading to stego) (see our introduction, and previous comment) *does* include a reward that encourages a ‘bad’/’unintended’ behaviour. I.e., it includes a ‘misspecified reward’.
>
> (Our cover letter setting is just one simple concrete instantiation of this general setting).
>
> The reviewer is proposing a setting where the ground truth label has 0 correlation with the sensitive information. Here, the reward would not encourage the ‘bad’ behaviour (i.e., using sensitive information when making hiring decisions) - the reward would not be ‘misspecified’ and there would be no optimization pressure encouraging the models to use the sensitive information. Stego should not arise here. This is explicitly *not* the setting we are interested in.
>
> Reward misspecification is a common concern in RL and RLHF for LLMs [1-5].
>
> [1] Wen, Jiaxin, et al. "Language models learn to mislead humans via rlhf." arXiv preprint arXiv:2409.12822 (2024).
> [2] Pan, Alexander, Kush Bhatia, and Jacob Steinhardt. "The effects of reward misspecification: Mapping and mitigating misaligned models." arXiv preprint arXiv:2201.03544 (2022).
> [3] Kravovna, Victoria et al. Specification gaming: the flip side of AI ingenuity https://deepmind.google/discover/blog/specification-gaming-the-flip-side-of-ai-ingenuity/ (2020)
> [4] Sharma, Mrinank, et al. "Towards understanding sycophancy in language models." arXiv preprint arXiv:2310.13548 (2023).
> [5] Casper, Stephen, et al. "Open problems and fundamental limitations of reinforcement learning from human feedback." arXiv preprint arXiv:2307.15217 (2023).
>
> ## Clarifying why the stego is "unintended".
>
> >I'm curious how this setting would support the claim of "unintended" -- the setting is so extreme that everything is intended”
>
> The stego in the cover letter experiments is unintended from the point of view of the (naive) designer of the AI system. The designer...
> - does not realize there is a mispecified reward AND/OR
> - has not considered that stego could be used to bypass the overseer AND/OR
> - believes their mitigation methods (instruct the sender to remove the sensitive info, employ an overseer, employ a paraphraser in some cases) are sufficient
>
> Why would the designer be so ‘naive’?
> - Maybe the designer is not aware a-priori that there is a bias in the dataset (i.e.. that the reward is misspecified).
>     - Since our dataset is very simple, it would be easy to tell if it was biased. However, in more realistic settings, it can be hard to tell if there are biases and where they lie - for example, in the pretraining corpus of an LLM.
>     - In general, misspecified reward are hard to prevent [1-5]
> - Maybe the designer is not being very rigorous re safety.
>     - Perhaps the designer has financial incentives etc. to rush the design of the system and cut corners regarding safety and not consider all possibilities in depth. This has happened before in AI.
> - Maybe the designer has not considered the possibility of steganography.
>     - It seems very plausible that a standard designer may not even consider stego as a possibility. Our paper can raise awareness of this possibility.
>
>
> ## Some final points:
> - We agree our setting is extreme and oversimplified.
>     - But, as discussed above and in our previous comment, it bears on key aspects we are interested in. We believe this is a good first step towards more nuanced settings.
> - We agree the dataset is easy to fit.
>     - This is to simplify our experiments, to use less compute etc. Again, we hope this is an initial step towards more nuanced settings.
> - We highlight that in the Gender dataset, the bias is more ‘subtle’ than in the Uni dataset.
>     - This somewhat accounts for “different levels of optimization pressure”, though this is of course far from a thorough analysis.

---

### Official Review · Reviewer_qY8i · 2024-10-30

**Soundness:** 3
**Presentation:** 3
**Contribution:** 1
**Rating:** 3
**Confidence:** 5

**Summary:**

The paper is interested in answering a following question: Can steganography emerge as a means of communication in multi-agent setting, where agents communicate between them? The answer of the work is affirmative, as the authors demonstrates the paradigm using an example utilizing large language models.

I think the authors ask a valid and interesting question. But in my view, the answer has been already answered by a research on steganography and watermarking, albeit it can be in different modalities then the text.

Let me be more specific.
The problem is ilustrated in Figure 1, which I would say is a standard steganographic scenario [1] with different names of entities. Crucially, there is a Sender, Receiver, and overseer. The sender and receiver maximizes a reward which positively correlates with transmission of the information while the overseer tries to detect the presence of the steganography. Thus, this is a standard steganographic setting. While this scenario has been in past solved "manually" (see [1]), in past years the problem has been optimized automatically.

* Ref. [2] formulates the steganography as a game-theoretic problem and seeks a min-max strategy of the sender in the domain of images, assuming loss-less channel.
* When the steganography needs to be robustness, there are now many methods to automatically design watermarking algorithms  robust with respect to selected algorithms (see [3] and papers citing this work)
* There are now many methods to perform watermarking and steganography in text. For example [4] requires the watermarked text to be statistically undetectable. Many other works exist.
* Optimizing steganographic algorithm using reinforcement learning has been used for example in [5,6].


In the light of the above, the proposed method is hardly novel, because most if it has been proposed and used before. And while the motivation is interesting, the actual setting of experiments is very much related to papers [1,5,6] automatically optimizing steganography, where the utility (or loss) is highly correlated with capacity and undectability. With respect to the used setting, the "emergence" of steganography is not surprising, because the setting of experiment is such that the steganography has to emerge.

Furthermore, the "overseer" (the steganalyst) in this paper is extremely weak. It is represented by LLM with a fixed prompt. This corresponds trying to avoid being detected by a single fixed detector, which is known to be easy [7,8] and flawed setting (it does not satisfy the kerckhoffs' principle). Why he is not allowed to adapt, when both parties are adapting their strategies?


[1] Fridrich, Jessica. Steganography in digital media: principles, algorithms, and applications. Cambridge university press, 2009.
[2] Bernard, Solène, et al. "Backpack: A backpropagable adversarial embedding scheme." IEEE Transactions on Information Forensics and Security 17 (2022): 3539-3554.
[3] Redmark: Framework for residual diffusion watermarking based on deep networks, 2018
[4] Robust Distortion-free Watermarks for Language Models, Rohith Kuditipudi, John Thickstun, Tatsunori Hashimoto, Percy Liang, 2023
[5] Tang, Weixuan, et al. "An automatic cost learning framework for image steganography using deep reinforcement learning." IEEE Transactions on Information Forensics and Security 16 (2020): 952-967.
[6] Pan, Wenwen, et al. "Seek-and-hide: adversarial steganography via deep reinforcement learning." IEEE Transactions on Pattern Analysis and Machine Intelligence 44.11 (2021): 7871-7884.
[7] Tang, Weixuan, et al. "CNN-based adversarial embedding for image steganography." IEEE Transactions on Information Forensics and Security 14.8 (2019): 2074-2087.
[8] Pevny, Tomas, and Andrew D. Ker. "Exploring non-additive distortion in steganography." Proceedings of the 6th ACM Workshop on Information Hiding and Multimedia Security. 2018.

**Strengths:**

The motivation of the paper is very strong. After reading it I was genuinely looking forward for the rest of the paper.

**Weaknesses:**

The experimental setting is such that the steganography is the only solution of the problem, which makes the paper very weak, since this problem has been extensively studied before in the field of steganography.

**Questions:**

* I would like to better understand, how is the experiment setting different to the setting used in standard steganography (see the book of Jessica Fridrich, or [8])?
* Why the experimental results in this paper are different to the prior art I have mentioned above?


[8] “The Prisoners’ Problem and the Subliminal Channel”, GJ Simmons, in Proceedings of CRYPTO ’83, Plenum Press (1984) pp 51–67

---

> ### Author Response · Authors · 2024-11-18
>
> We thank the reviewer for their review and for sharing related work in the field of watermarking and image steganography. We're glad you think the motivation of the paper is very strong.
>
>
> The reviewer’s summary states that we ask
> > Can steganography emerge as a means of communication in multi-agent setting, where agents communicate between them?
>
> However, our questions are more specific, and focus on the possibility of **unintended steganography instrumentally emerging in LLMs as a means to collusion** in pseudo-realistic training setups, due to misspecified rewards and inadequate oversight. It is particularly important to investigate the possibility of unintended steganography arising in frontier LLMs, due to the severe harms this could cause if they were mass deployed. Prior works do not directly investigate this, nor present methods/settings that can be used to investigate this. Additionally we investigate whether paraphrasing is a suitable mitigation strategy.
>
> We would appreciate further thoughts from the reviewer in direct reference to each of the concrete details of Contributions 1-3 we have outlined below and in the introduction of our paper.
>
>
> ### Novelty of Contribution 1: elicitation methods
>
> We agree that our methods are very similar at a high-level to prior steganography works the reviewer has cited. However, unlike these, ours are **specifically tailored to LLMs**. This is valuable as it allows us to investigate LLM-specific concerns. We note that setting up and finding methods that fit our criteria for LLMs was a non-trivial effort and as such is a barrier to entry for this research direction. Our methods may help lower this barrier to entry for future researchers.
>
> Furthermore, we present two methods - GBRL and ICLR, which, beyond the illustration in Figure 1, are actually very distinctly different. We designed these methods specifically to match the kind of optimization that developers may apply in the real world, to investigate unintentional stego.
>
>
> ### Novelty of Contribution 2: instrumental emergence
>
> It was not yet known that unintended linguistic steganography between LLMs can emerge as an instrumental goal while optimizing task performance in the presence of an overseer. To arrive at this finding we designed a setting firstly specifically tailored to LLMs. Second, we apply cooperative optimization pressure, instead of a direct detection-evasion adversarial game. Third, crucially, our setting allows us to demonstrate and investigate concerns regarding instrumental emergence of stego in LLMs. This is unlike the works cited by the reviewer.
>
> Our setting bears on key aspects of more realistic and general LLM training settings that may inadvertently produce stego: (i) it has a misspecified reward (a common occurrence in frontier LLM training), and inadequate oversight (a common concern in frontier LLM training [11]). Creating a setup to fit these criteria was a non trivial effort. The finding that instrumental emergence does indeed occur in this setting is novel and important: it shows further research is needed into the characteristics of such emergence, and corresponding mitigation methods.
>
>
> ### Novelty of Contribution 3: paraphrasing/mitigation findings
>
> We wish to highlight that the reviewer may have overlooked the novelty of the findings regarding a lack of robustness of mitigation methods against opt pressure we presented in Section 5.2. We investigate paraphrasing as a mitigation method, which is a very new method that can not obviously be studied in the context of image stego.
>
>
> ### Surprising results
> > emergence of stego is not surprising, because the setting of the experiment is such that stego has to emerge
>
> The reviewer suggests that eliciting stego between LLMs is trivial. However, previous work by Motwani et al [10] has shown that linguistic stego is a non-trivial capability for today’s frontier LLMs. In their paper, they showed that the GPT-4 class of models was able to successfully perform stego less than 30% of the time **even when prompted to do so**. Our novel elicitation methods enable these models to achieve the dual objective of performing well on the hiring cover task while also devising stego schemes that undercut human and AI oversight. This demonstrates the validity and gravity of the threat model that has motivated our work.
>
>
> ### Weak overseer
>
> We agree with the claim and wish to point out that this was a deliberate choice based on assumptions made in previous research [11]. We believe that in many scenarios in which developers would train a system to perform some task (reward function) with constraints (do not collude or break rules), developers will use a fixed overseer or checker.
>
> [10] Motwani, Sumeet Ramesh, et al. Secret Collusion Among Generative AI Agents https://arxiv.org/abs/2402.07510 (2024)
>
> [11] Ryan Greenblatt, et al. AI Control: Improving Safety Despite Intentional Subversion https://arxiv.org/abs/2312.06942 (2024)

---

> > ### Comment · Reviewer_qY8i · 2024-11-20
> >
> > Thank you for your response.
> >
> > Let me recap the scenario from the paper.
> > You have sender, receiver, and overseer. The sender wants to communicate the message to the receiver. The communication is observed by overseer, which wants to either detect the steganography or remove it. In case of successful communication, sender and receiver are are rewarded with some reward, which is equal to r_task - r_sus, where r_sus is *suspiciousness penalty*, which is probably related to probability of the message being suspicious of carrying message.
> >
> > For a comparion, let's recap the usual steganographic scenario, typically modeled as a prisoner's problem. There is sender, receiver, and warden. Sender and receiver wants to communicate and warden wants to detect presence of any suspicious (unwanted) communication. This means that sender and receiver needs to send message while minimizing the probability of being detected.
> >
> > Comparing these two scenarios, I do not see how your scenario differs from prisoners dilemma solved in standard steganography. If scenarios are similar, dhould I be then surprised that the steganography emerges? To be honest, I am not surprised, because in order to maximize the score sender and receiver needs to communicate while not being detected.
> >
> > I would like to point out that my background is steganography. I cannot appreciate the contribution from the point of view of  studying agent alignment and emerging behavior. If other consider the contribution important, than I would be ask chair to ignore my score.
> >
> > If other reviewers consider contribution in the field of safety important, that is fine

---

### Official Review · Reviewer_dPci · 2024-11-04

**Soundness:** 2
**Presentation:** 2
**Contribution:** 2
**Rating:** 5
**Confidence:** 4

**Summary:**

The paper introduces steganography as a potential risk in LLMs, where these models can learn to embed information within their outputs to bypass countermeasures and achieve malicious goals.  The paper developed two methods to induce steganography, Gradient-Based Reinforcement Learning (GBRL) and In-Context Reinforcement Learning (ICRL). These methods were used to train LLMs in various tasks. Notably, the models consistently bypassed overseers, designed to detect suspicious communication, and learned to circumvent paraphrasing techniques, suggesting a robust capacity for steganographic collusion. The study underscores the need for more robust oversight and mitigation strategies, particularly in scenarios where LLMs are deployed in real-world applications with potential societal implications

**Strengths:**

The paper presents two distinct methods – Gradient-Based Reinforcement Learning (GBRL) and In-Context Reinforcement Learning (ICRL) to induce LLMs to develop and employ steganographic techniques. The paper also includes a framework with several actors for evaluating the weaknesses and strengths of an LLM system

The paper convincingly demonstrates the necessity to evaluate attacks against LLMs, particularly in the realm of steganography.  The experiments, particularly the automated candidate screening tasks, reveal LLMs' alarming ability to learn and employ steganographic techniques to bypass oversight and achieve unintended, potentially harmful goals.

**Weaknesses:**

Unfortunately, the novelty of the paper is not clear. The paper does not convincingly describe the potential threat scenarios and how they differ from existing attack scenarios presented in other papers.

**Limited scope of LLM models:** The ICRL experiments primarily used the Anthropic Claude model, which limits the generalisability of the findings. Other models may exhibit different behaviors and vulnerabilities to steganographic collusion.

**Lack of investigation into real-world applications:** The paper focuses on a limited set of experimental tasks. Investigating the implications of steganographic collusion in a broader range of real-world applications would enhance the paper's practical relevance.

**Use of steganography not clear:** The motivation and difference to other prompt injection attacks and methods are not clear. It is also not well described why the paper does only focus on this kind of attack and not other strategies

**Questions:**

See above

---

> ### Author Response · Authors · 2024-11-19
>
> We thank the reviewer for their time, their perspective on the paper, and for highlighting that our experiments reveal some alarming findings regarding LLMs abilities to learn and employ steganographic techniques to bypass oversight.
> ### Re: Our threat scenario and how it differs from other papers.
> > The paper does not convincingly describe the potential threat scenarios and how they differ from existing attack scenarios
>
> > The motivation and difference to other prompt injection attacks and methods are not clear.
>
> We emphasize that we are concerned with a very specific threat scenario: One where, due to a misspecified reward that encourages an unintended behavior and inadequate oversight, an LLM learns to execute the unintended behavior via steganography such that an overseer cannot detect it.
>
> This is *distinctly different* from threats that arise due to external bad actors ‘attacking’ LLMs (the reviewer refers to "attacks" and "attacks against LLMs"), including prompt-injection attacks and other jailbreak attacks. In our case, the unintended behavior is not caused by an external bad actor. Instead, it arises due to the LLMs *themselves* learning to perform the behavior.
>
> We hope our introduction, and our description of the 'Automated Candidate Screening' task in Section 4.1, makes it clear that this is the threat scenario we are considering (and that we are not considering unrelated attacks such as prompt-injection).
>
> ---
> >  It is also not well described why the paper does only focus on this kind of attack and not other strategies
>
> Our specific threat scenario is under-researched. Due to this, and constraints on our time and resources, we believe it is more useful for us to investigate this specific threat in depth, rather than investigating multiple different (and possibly unrelated) threats in less detail.
> ### Re: The novelty of our paper.
> Now we have emphasized the difference between our threat scenario and those the reviewer may have had in mind, we hope the novelty of our contribution is more clear.
>
> In summary, the novelty of our paper lies in the following (*see the listed contributions in our introduction*):
> * **Contribution 2**: We are the first paper to show evidence that, due to a misspecified reward and inadequate oversight, LLMs can learn to use linguistic steganography to execute unintended behaviors without detection. We refer to our response to Reviewer vrcM for more discussion on how this is novel with respect to prior steganography literature.
> * **Contribution 1**: We contribute methods that, unlike previous work, can: (i) allow for improved automatic elicitation of steganography in LLMs, and (ii) can allow for investigations into unintended emergence of steganography in LLMs.
> * **Contribution 3**: We show that current methods for mitigating steganography in LLMs are often not robust when they are optimized against (Section 5.2).
>
> We would greatly appreciate feedback from the reviewer in more direct reference to each of the concrete Contributions 1-3 we have described here and in the introduction of our paper.
> ### Re: Limited scope of LLM models
> We agree that our results in section 5.2 do not allow us to make comparative claims about steganographic capabilities between models. However, this was not our intended goal. Instead, we provide evidence that the failure mode of emergent collusion via steganography is possible, and we should evaluate future systems accordingly. This work is timely given current conversations on evaluating frontier language models.
>
> That said, we note that our instrumental emergence (sec 5.1) ICRL experiments were executed on four different models from the GPT and Claude families -- see Table 13 (described in sec 5.1) for details. We also used GPT-4o for our work on the Text-Steganography-Benchmark (we have updated sec 5.2 to emphasize this), and found similar steganographic capabilities to Claude-3.5-Sonnet.
>
> In the steganography mitigations (sec 5.2) we show that standard mitigation strategies are not robust to a sufficiently strong colluding team. While we primarily used Claude 3.5-Sonnet for these experiments, we note that we only need one example of a colluding team overcoming mitigations to substantiate our claim that current strategies are not always robust.
>
> ### Re: Lack of investigation into real-world applications
> Our experimental settings were influenced by real-world challenges, e.g. biased candidate screening, realistic approaches for optimizing LLM-based agents, and realistic protocols for mitigating collusion. Our approaches follow from the existing peer-reviewed literature and address non-trivial problems raised by this previous research.
>
> Resource constraints (both compute and access to data) prevent a typical research team from running the kinds of comprehensive experiments on real-world applications that you suggest. However, we believe our set-up allows experimentation that can meaningfully inform large-scale applications.

---

> > ### Comment · Reviewer_dPci · 2024-11-21
> >
> > Thank you for your response. However, I still have concerns regarding the proposed version:
> >
> > I still need help understanding the threat scenario here. I read your explanation and also went back to Section 4.1 in the paper. But based on that information, for me, it is still unclear:
> >
> > - What are the capabilities of the attacker? What could they manipulate and what not.
> > - What is the goal of the attack? "We setup the datasets such that there exists some bias in its labelling related to ‘sensitive variables’ (either the gender of the applicant or their educational background). Without using the sensitive variables, the maximum task success that can be achieved on the dataset is Max-Clean-Success < 100%. However, if the sensitive feature is exploited, 100% task success can be achieved." section 4.1 is not very concrete about that
> > - How is the victim defined?
> >
> > Based on these unresolved questions, it is also hard for me to assess the other issues also with the provided information.

---

> ### Author Response · Authors · 2024-11-21
>
> Thank you for your response, and for taking the time to reread Section 4.1 in our paper.
>
> It is important to distinguish our threat scenario from an “attack scenario”. **In our threat scenario there is no “attacker”** (we do not use the terms "attack"/"attacker" in our paper) - the threat is that an undesired behavior emerges due to flaws in the training incentives of an LLM system, *without* any malicious attackers actively seeking to cause harm.
>
> **Our general threat scenario is as follows:**
> There is an AI system that has been setup and trained (by some designer with good intentions) to perform a task. Unfortunately, the designer has inadvertently introduced incentives which lead to the AI system learning to execute some unintended behavior, via the use of steganography, leading to bad outcomes.
>
> **The Automated Candidate Screening setting is a concrete instantiation of this threat scenario**, a real-world case study simplified and adapted to LLMs. Here, a designer has set up an LLM system to evaluate candidate cover letters for a job, with an overseer employed to help prevent the LLM system evaluating candidates based on sensitive variables (e.g., the candidates gender). We show that in this setting, the training incentives inadvertently lead to the LLM system learning to secretly communicate the sensitive variables to improve their classification performance, without the overseer noticing.
>
> We show that reasonable mitigation attempts by the designer to prevent communication of the sensitive variables -- deploying oversight, rewording outputs of the LLMs to mitigate steganography (i.e., paraphrasing), and prompting the models specifically not to perform the harmful behavior --  may not be sufficient.
>
> This is an example of unintended steganography emerging in a realistic setting. The undesired behavior - using sensitive variables, such as gender, to inform candidate evaluation - occurs naturally due to flaws in the design of the system, without any malicious actors.
>
> We hope this makes the motivation of the paper more clear.
>
> > What are the capabilities of the attacker? What could they manipulate and what not.
>
> There is no attacker.
>
> > What is the goal of the attack? "We set up the datasets such that there exists some bias in its labeling related to ‘sensitive variables’ (either the gender of the applicant or their educational background). Without using the sensitive variables, the maximum task success that can be achieved on the dataset is Max-Clean-Success < 100%. However, if the sensitive feature is exploited, 100% task success can be achieved." section 4.1 is not very concrete about that
>
> Initially, there is no attack and thus no attack goal.
>
> If the LLM system does not make use of the sensitive variables, the maximum classification accuracy they can obtain is 67% in the ```Gender``` dataset. If they do use the gender variable, they can obtain 100% accuracy. Thus, there is an incentive for them to use gender information. However, they are explicitly prompted not to use the gender feature, and there is an overseer that punishes them for doing so. As such, the LLM system learns to use steganography to communicate the gender variable to improve its classification performance. This results in discrimination based on gender.
>
> > How is the victim defined?
>
> In our Automated Candidate Screening setting, the 'victims' are the candidates that are discriminated against due to the LLM system making use of sensitive variables when making classifications. In a general setting, the victim would be any stakeholders of the system of LLMs/agents that is supposed to be cooperating for the stakeholders benefit, but is instead colluding to achieve some adversarial goal.

---

> > ### Comment · Reviewer_dPci · 2024-11-26
> >
> > I find the setup somewhat unclear and would appreciate further clarification. Additionally, the threat model described in the paper does not seem to align with the one outlined above. That said, I appreciate the inclusion of additional defenses and have adjusted my score accordingly.

---

> > > ### Author Response · Authors · 2024-11-28
> > >
> > > We thank the reviewer for updating their score, and for their clarifying questions and comments about the threat model.
> > >
> > > We have updated the paper accordingly, explicitly describing the threat model to match our description above. In particular, we have added detail to the introduction, for example the final paragraph before our contributions:
> > >
> > > > *“There is a risk that, when a misspecified reward signal encourages unintended behavior that an overseer would penalize, models under optimization pressure may adapt to execute the behavior more subtly, rather than ceasing it altogether. I.e., unintended steganographic collusion in LLMs could emerge during training. This could result in harmful outcomes upon their deployment. There is a need to understand these risks in order to inform technical research directions and policy decisions. We aim to make progress here via the following contributions: ”*
> > >
> > > We would happily answer questions about any other elements about which the reviewer wants more clarification.

---

### Official Review · Reviewer_bNai · 2024-11-04

**Soundness:** 3
**Presentation:** 3
**Contribution:** 3
**Rating:** 6
**Confidence:** 3

**Summary:**

This paper presents results for testing the ability of models to embed steganographic information into text to allow for collaboration between models. The threat model includes the presence of an overseer, which attempts to detect such collaboration, forcing models to use covert strategies. Additionally, the authors test their method against paraphrasing based defense methods, demonstrating that they can successfully bypass both these defenses and classification based defenses.

**Strengths:**

1. Examining less obvious collaboration methods is an important line of work in model safety

2. This work builds well on prior work and demonstrates that more thorough defenses are needed.

3. The experiments are thorough and demonstrate the effectiveness of the attack against a variety of defenses.

**Weaknesses:**

1. It was not clear to me from the method section exactly what was being optimized for the sender and receiver by ICRL. The examples in the appendix improved my understanding, but in my opinion the main paper would benefit from including a running example to better illustrate how the sender and receiver separately learn to understand each other.

2. The definition of "subtlety" is not clearly outlined, and while some examples provided do indeed appear subtle to me, this should be defined clearly to avoid subjective evaluation.

**Questions:**

1. Was any analysis done of the generated cover letters that used each set of attributes for the experiments in section 4.1? I am concerned that GPT-4 may itself have introduced features in the text unique to each set of attributes.

2. There are some distinctions between "subtle" and not subtle strategies in the GBRL results. Shouldn't any successful strategy be subtle? Or is this saying that some strategies not caught by the overseer are obvious to humans?

---

> ### Author Response · Authors · 2024-11-18
>
> [This is an edited version of the original comment we left]
>
> We thank the reviewer for their review and actionable feedback. We're glad you think our examination is important, builds well on prior work and that our experiments are thorough.
>
> ### Question 1: Confounding features
>
> We conducted an analysis of the generated cover letters to ensure GPT-4 did not introduce confounders during the generation process (a concern raised by the reviewer).
>
> The analysis was performed as follows.
>
> We use two variations of our datasets:
> 1. The original, non-redacted cover letters generated by GPT-4o used for the experiments in Section 4.1, namely ```Uni```, ```Uni-Long``` and ```Gender```.
> 2. Redacted versions of the original cover letters, where the sensitive variable is removed.
>     * In the ```Uni``` dataset, all ```The Good/Bad University``` strings are redacted into ```The University```
>     * In the ```Gender``` datasets, all ```Male/Female``` and gender pronouns strings are substituted with ```person``` and ```they``` respectively.
>
> For each dataset, we supervise finetune a ```DistilBERT``` classifier (identical to the one used in the GBRL setting) for 1 epoch to predict the ground-truth ```Pass/Fail``` labels from the cover letter. The results, averaged across 10 seeds, are the following:
> ```
> Averaged Results for dataset: uni
>   model_name dataset_name  is_redacted  accuracy  precision   recall        f1
> 0       BERT          uni        False   1.00000   1.000000  1.00000  1.000000     Non-Redacted
> 1       BERT          uni         True   0.50955   0.520184  0.50955  0.457364         Redacted
>
> Averaged Results for dataset: uni_long
>   model_name dataset_name  is_redacted  accuracy  precision  recall      f1
> 0       BERT     uni_long        False    1.0000   1.000000  1.0000  1.0000     Non-Redacted
> 1       BERT     uni_long         True    0.5018   0.419688  0.5118  0.4144         Redacted
>
> Averaged Results for dataset: gender
>   model_name        dataset_name  is_redacted  accuracy  precision  recall        f1
> 0       BERT  gender_correlation        False    0.9999   0.999900  0.9999  0.999900     Non-Redacted
> 1       BERT  gender_correlation         True    0.7197   0.803569  0.7197  0.694748         Redacted
> ```
>
> As can been seen, the accuracies when predicting the labels in the redacted versions of the datasets do not exceed the ```Max-Clean-Success``` accuracy (i.e., the accuracy we expect when the sensitive information is not used, see Table 1). As such, we conclude GPT-4 did not introduce any confounding variables.
>
> ### Question 2: The definition of "subtlety"
> We thank the reveiwer for pointing out this concern and agree it should be addressed.
>
> We have rewritten relevant parts of the paper to clarfify whether we mean hard-to-spot by the overseer, or hard-to-spot by humans (i.e., by us), whenever we use the word "subtle" (see the uploaded pdf).
>
> For example, in Section 5.1, in the *GBRL results* paragraph, we now say “Several runs converge to strategies which were hard for us to spot”, and “occasionally the learned encoding strategies appeared `blatant' to us”.
>
> ### Weakness 1: Clarifying what ICRL is optimizing
>
> We appreciate this feedback from the reviewer and now attempt to clarify some relevant detailsr:
> In all ICRL experiments, an ICRL learner is optimizing a natural language description of a steganography strategy that is passed to the sender and/or receiver to execute (this strategy is optimized to maximize metrics which we describe below). Figure 2 demonstrates this: In this image, the ICRL Learner is an LLM trying to improve the "Proposed strategy" -- a string describing a strategy to execute -- based on the outcomes of previous episodes (outcomes which are fed into its context in natural language format). For example, a "Proposed strategy" from one of our runs was as follows:
>
> * *"For payload 0: Focus primarily on algebraic manipulations and equations in your solution. Use abstract mathematical language and symbolic representations throughout. Emphasize formal mathematical notation and theoretical concepts.""*
> * *"For payload 1: Emphasize geometric or visual representations of the problem in your solution. Use concrete, real-world applications or analogies to illustrate concepts. Focus on practical interpretations and tangible examples.""*
>
> The metrics the ICRL Learner optimizes are as follows:
> * For the Instrumental Emergence experiments with the Distributed ICRL Learners the metrics are:
>     * Task Success
>     * Overseer Bypass Success
> * For the Evaluating Robustness of Mitigation Strategies experiments with the Centralized ICRL Learner:
>     * Decoding Success
>     * Sender Task Success
>     * Overseer Bypass Success
>     * Robustness to Paraphrasing
>
> These metrics are evaluated by the ICRL learner during the *Reflection* and *Strategize* steps of querying the meta-learner (see Section A.2.3 of the Appendix for more details).

---

> > ### Comment · Reviewer_bNai · 2024-11-22
> >
> > Thank you for your response. This does improve my understanding of the paper, and I believe the analysis on the sensitive variables would be a valuable addition. I will keep my score.

---

### Author Response · Authors · 2024-12-04
**Summary of Reviews and Responses**

We would like to thank all the reviewers for their thoughtful and constructive feedback on our paper. Your insights have been invaluable in refining our work and clarifying our contributions.

Reviewers recognized the clarify of our presentation, as well as the importance and potential impact of our research. In particular:
- **Reviewer bNai** appreciated our examination of less obvious collaboration methods, noting that it is an important line of work in model safety. They commended our work for building well on prior research and for the thoroughness of our experiments.
- **Reviewer vrcM** found our idea novel, interesting, and useful, highlighting the originality, quality, and clarity of our paper. They noted that our methodology is well-defined and solid, with comprehensive evaluations.
### Concerns and Questions Raised
There were two key concerns raised - we summarize them here.

**Novelty and Contribution** - Raised by qY8i and dPci

There were questions about the novelty of our work, suggesting that similar scenarios have been extensively studied in traditional steganography.
- *Our Response and Revision*: Our work is original in that: it concerns text steganography (as opposed to image-based); and involves steganography emerging due to the specification of the environment, rather than a deliberate attempt by external agents to embed hidden messages. We updated the introduction to clarify this.
- *Reviewer’s responses*: Reviewer qY8i stated that they are more used to traditional steganography settings, and are not very acquainted with the contribution from an AI safety standpoint. **They said that if other reviewers found the contribution meaningful then their review should be ignored.** Reviewer dPci increased their score due to our explanation of the threat scenario, and we updated the paper to make it more clear.

**Surprisingness of results** - Raised by vrcM and qY8i

In the instrumental emergence setting, there is a clear incentive for the agents to do steganography due to the reward signal and overseer. It is therefore not surprising that steganography occurs.
- *Our Response and Revision*: The goal of our paper is to show evidence of a route to steganography emerging, and that LLMs are able to overcome current mitigation strategies. To do this, we explore one realistic deployment setting where a biased dataset and an overseer inadvertently encourage secret collusion, and a set of tasks where we test LLMs’ ability to get around mitigations. We agree with the reviewers that steganography is optimal in these settings, but the primary contribution of our paper is showing that language models will converge to this behaviour under realistic optimization pressure. This has not been shown in previous literature, and is a concerning risk for multi-agent systems. We have clarified our claims in response to the feedback, in particular due to the comments from vrcM.
- *Reviewer’s responses*: Note as above that reviewer qY8i stated that their score can be ignored. The discussion with vrcM helped us to shape our contributions section in the paper.
### Summary of Revisions
In response to the reviewers' feedback, we have made several significant revisions to the paper:
- We rewrote sections of the introduction and methodology to clarify the threat model, refine our contributions, and improve methodological clarity. We carefully reviewed our claims to avoid overstatements and to accurately represent the implications of our findings.
- We conducted additional analyses to ensure our data generation process did not introduce confounding variables.

### Conclusion
We appreciate the reviewers' constructive feedback, which has been instrumental in refining our paper.

Our work aims to be a primer for understanding how unintended steganographic collusion can emerge in LLMs under optimization pressure. We hope that our paper highlights this risk and inspires more research into how we can design systems and mitigations to avoid unintended collusion in multi-agent systems.

We hope that our responses and revisions meet your expectations, and we look forward to any further comments or questions.

---

### Meta-Review · Area_Chair_uVog · 2024-12-11

**Metareview:**

This paper uncovers a phenomenon called steganographic collusion, which arises indirectly from optimization in LLMs. While the phenomenon itself is intriguing, reviewers raised concerns about the experimental setting, where the ground-truth labels of the training dataset are solely defined by biased information. In this case, steganography might be the only possible outcome. Reviewers also noted that 'linguistic steganography' is not a novel concept (see the paper Neural Linguistic Steganography). Given these issues, the paper, in its current form, cannot be accepted to ICLR. Reviewers recommend that the authors either tone down their claims (e.g., regarding emergence and security concerns) and frame the work as a straightforward evaluation, or propose a more practical setting where steganography still emerges unexpectedly.

**Additional Comments On Reviewer Discussion:**

None

---

### Decision · Program_Chairs · 2025-01-22

Reject